# Salorno—Dos de la Forca (Adige Valley, Northern Italy): A unique cremation site of the Late Bronze Age

Federica Crivellaro[1,2], Claudio Cavazzuti[3,4], Francesca Candilio[5], Alfredo Coppa[2,6,7], Umberto Tecchiati[8]*

**1** Turkana Basin Institute, Stony Brook University, Stony Brook, New York, United States of America, **2** Dipartimento di Biologia Ambientale, Università di Roma "La Sapienza", Roma, Italy, **3** Dipartimento di Storia Culture Civiltà, Università degli Studi di Bologna, Bologna, Italy, **4** Department of Archaeology, Durham University, Durham, United Kingdom, **5** Dipartimento di Bioarcheologia, Museo delle Civiltà, Rome, Italy, **6** Department of Evolutionary Anthropology, University of Vienna, Vienna, Austria, **7** Department of Genetics, Harvard Medical School, Boston, Massachusetts, United States of America, **8** Dipartimento di Beni Culturali e Ambientali, PrEcLab - Laboratorio di Preistoria, Protostoria ed Ecologia Preistorica, Università degli Studi di Milano, Milano, Italy

* umberto.tecchiati@unimi.it

**Data Availability Statement:** All relevant data are within the paper and its Supporting information files.

## Abstract

The archaeological site of Salorno—Dos de la Forca (Bozen, Alto Adige) provides one of the rarest and most significant documentations of cremated human remains preserved from an ancient cremation platform (*ustrinum*). The pyre area, located along the upper Adige valley, is dated to the Late Bronze Age (*ca.* 1150–950 BCE) and has yielded an unprecedented quantity of cremated human remains (about 63.5 kg), along with burnt animal bone fragments, shards of pottery, and other grave goods made in bronze and animal bone/antler. This study focuses on the bioanthropological analysis of the human remains and discusses the formation of the unusual burnt deposits at Salorno through comparisons with modern practices and protohistoric and contemporaneous archaeological deposits. The patterning of bone fragmentation and commingling was investigated using spatial data recorded during excavation which, along with the bioanthropological and archaeological data, are used to model and test two hypotheses: Salorno—Dos de la Forca would be the result of A) repeated primary cremations left *in situ*; or B) of residual material remaining after select elements were removed for internment in urns or burials to unknown depositional sites. By modelling bone weight and demographic data borrowed from regional affine contexts, the authors suggest that this cremation site may have been used over several generations by a small community–perhaps a local elite. With a quantity of human remains that exceeds that of any other coeval contexts interpreted as *ustrina*, Salorno may be the product of a complex series of rituals in which the human cremains did not receive individual burial, but were left *in situ*, in a collective/communal place of primary combustion, defining an area of repeated funeral ceremonies involving offerings and libations across a few generations. This would represent a new typological and functional category that adds to the variability of mortuary customs at the end of the Bronze Age in the

**Funding:** FC, 5.4/14.05.07/010/2915, Provincia Autonoma di Bolzano (IT); UT, 0002920/22, Università degli Studi di Milano (IT). The funders had no role in study design, data collection and analysis, decision to publish, or preparation of the manuscript.

**Competing interests:** The authors have declared that no competing interests exist.

Alpine are, at a time in which "globalising" social trends may have stimulated the definition of more private identities.

## Introduction

Starting from the 2nd millennium BCE up to the Roman period, human cremations are extensively practiced across the whole of the Italian peninsula, the Alpine area, and Central Europe. However, archaeological documentation of funeral pyres (or *ustrina*) is scarce and patchy [e.g. 1–7]. Such lack of evidence is not surprising if we think that ancient cremations were operated outdoors, by means of pyres that were seldom equipped with permanent or semi-permanent structures. As confirmed by a number of experimental archaeological studies, funeral pyres are extremely ephemeral in nature [1, 8–13]. Unless built with permanent or semi-permanent structures, it is difficult for pyre residuals to remain on the ground. Ashes and small pieces of charcoal could be left *in situ* and be easily dispersed by atmospheric agents. Due to such ephemerality in the cremation process, we only have a partial understanding of the rituals connected to the cremation of the dead – more to deal with the post-cremation treatment of the bones (the so-called *ossilegium*), than with the preliminary (preparation of the corpse) and central (cremation) phases of the funeral ceremony.

The majority of the Late Bronze Age cremations from Central Europe is broadly addressed as 'urnfield tradition' and is characterised by the use of urns to keep human cremains along with other burnt goods and offerings, normally deployed in cemeteries [14–16]. More rarely are bone remains buried in the ground without a container.

This study presents and discusses the finding of a unique case study in the Upper Adige Valley (Italy), characterised by unusual burnt deposits with an extraordinary quantity of human cremains along with associated archaeological remains. In particular, the authors examine two hypotheses for understanding whether Salorno—Dos de la Forca (abbreviated Salorno) may be interpreted as A) the result of repeated primary cremations left *in situ*, or B) the result of residual materials remaining after select elements were removed for internment in urns or burials to unknown depositional sites. Thanks to a particularly accurate work of documentation during excavation, and to an accurate analysis of the size, weight and spatial distribution of the cremains, the site of Salorno offers new insights into the variability of mortuary rituals in Central Europe during the Bronze Age.

### Protohistoric cremations between Europe, the Po plain and the Alps

The abundance of cremated human remains contained in urns and pits dated from the Middle Bronze Age (1450 BCE) through the Early Iron Age confirm that this ritual was by far the most popular practice in Europe during that time. Unfortunately, the contemporary archaeological record is less generous with evidence of sites where ancient cremations (*ustrina*) took place–resulting in lack of information concerning the different phases of the ritual.

The cremation ritual must have been a spectacular event *per se*–whereby the use of fire implied complex and expensive operations, ranging from the preparation of the corpse and of the pyre itself; to the urn and grave goods; to the sorting, collection, washing and deposition of the bones inside the urn (*ossilegium*); to libations in honour of the deceased during and after the pyre; to the final deposition in the necropolis [17, 18]. Each of these operations must have entailed a symbolic/religious value, confirmed by the continuity of this ritual over many centuries across the late 2nd and 1st millennium BCE in Northern Italy. In fact, the transition from

inhumation to the widespread cremation ritual in various European regions during the central and later phases of the Bronze Age has been interpreted as a moment of shift in the phenomenology or the aesthetics of death, of the memory of the ancestors, and of the relationship with the world and matter [15, 16, 19–21]–possibly sided by ecological, social, political, as well as sanitary reasons [22].

The widespread adoption of the cremation ritual during this period–which in the past was monolithically defined *Urnenfelderkultur* [15]–started out discontinuously and gradually took more regional connotations, due to syncretism with local traditions. The diffusion of the 'urnfield model', with hundreds or thousands of burials placed side by side, did not spread in a linear manner across Europe. From the place of origin in the Danubian plains (in the context of the Vatya tells, between the Danube and the Tisza around 2000 BCE), it expanded along preferential lines and network corridors. From the Middle Bronze Age 2/3 (Bronzezeit B2/C1; 1500–1450 BCE) we observe a precocious and massive adoption of the 'urnfield' among the lowland Terramare in the Po Valley, and groups from the open fields of the Balkan regions–between the Danube, the Sava and the Drava rivers (e.g. in the context of the Belegis 1 or Virovitica culture [16, 23, 24]. Conversely, the Alps, Istria and Karst, seem to linger in the adoption of this ritual. Starting from the Late Bronze Age, urn cremations appear also in the Alpine valleys, more often included in small burial clusters.

Due to such variability of the 'urnfield model' at regional level, the placement of Salorno is not straightforward, being located across the border between the Alpine and the Po lowlands cultures.

## The archaeological context of Salorno—Dos de la Forca

The site of Salorno—Dos de la Forca is located on the left river basin of the Adige river–about 30 kilometres south of Bozen (Alto Adige or South Tyrol) [25] (Fig 1). The locality has been thoroughly investigated, providing archaeological evidence starting from the Early Holocene [26]. However, a more systematic occupation of the area occurs across the late prehistory, and the protohistoric and Roman ages. The micro-environmental context of the site is that of a slope deposit at the base of rocky cliffs – here practically vertical – which make up the western flank of Monte Alto (Geiersberg, 1083 m a.s.l.) on the easternmost edge of the Adige alluvial plain (S1 Fig). These accumulations, due to the excellent quality of the debris of which they are made, have been intensively cultivated for the extraction of gravels and construction aggregates [25]. In the Late Bronze Age, Salorno sat on a natural bottleneck in the Adige Valley–not suitable for either agriculture or settlement. This might explain the choice of the site for funerary rituals, although other symbolic/ideological reasons connected to the proximity of the site to water are not to be excluded.

The site was discovered in 1986 in the context of a gravel quarry (Cava Girardi). In the following year, the Provincial Superintendence for Cultural Heritage of Bolzano undertook archaeological investigations in an area partially affected by mechanical excavators [27]. A stratigraphic excavation revealed a sub-circular feature (named 'US 11') of about 6 m in diameter (Fig 2), characterized by a strongly carbonaceous earthy compound – very rich in ceramic remains, minute burnt bone fragments, glass paste beads, and bronze and antler objects (S2 Fig) – which immediately pointed to an ancient activity as funeral pyre and connected area of funerary worship. Within US 11, two concentrations of ceramic fragments, up to 20 cm deep, were exposed (features US 14 and US 18), providing for several dozens of shattered vessels with sharp edges, in almost direct contact with each other, minutely fragmented as if repeatedly trampled (S3 Fig). Unlike the rest of US 11, no burnt bones were found among the potsherds, suggesting that the cremation platform had specific zones for different ritual activities.

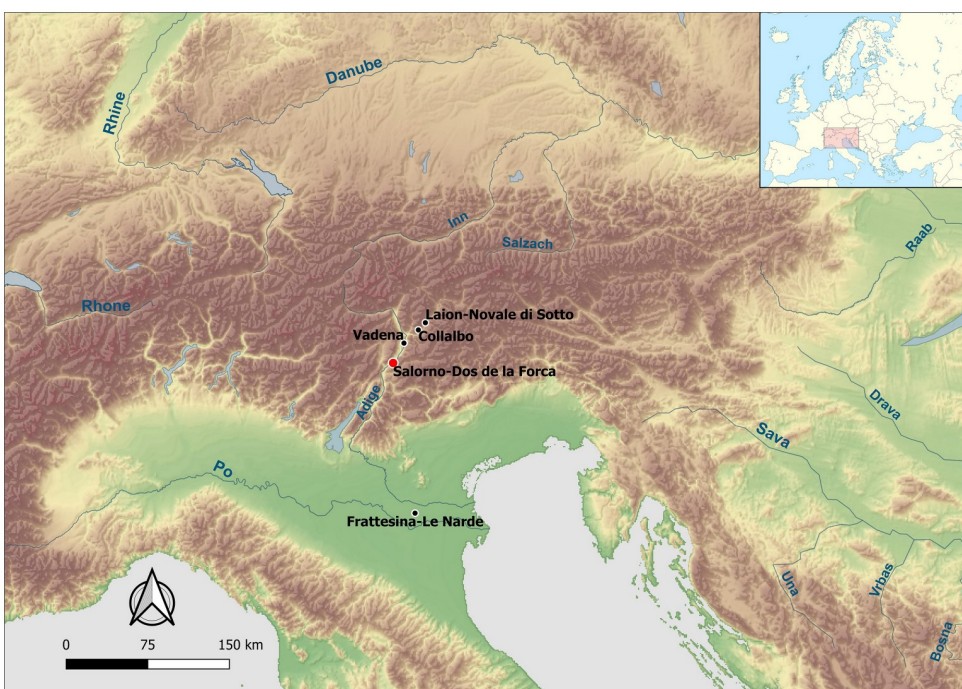

**Fig 1. Geographic location of Salorno and other sites mentioned in the text.** The map is constructed using "Natural Earth. Free vector and raster map data @ naturalearthdata.com" available at https://www.naturalearthdata.com/downloads/10m-raster-data/.

The south-west margin of the area was delimited by a large, squared boulder, cut by a wide and deep crack in its length, and with an almost flat surface. Although it is impossible to establish whether the boulder served for ritual purposes, this should not be excluded. All the archaeological materials point to a chronology in the Italian Final Bronze Age, namely 1150–950 BCE [25].

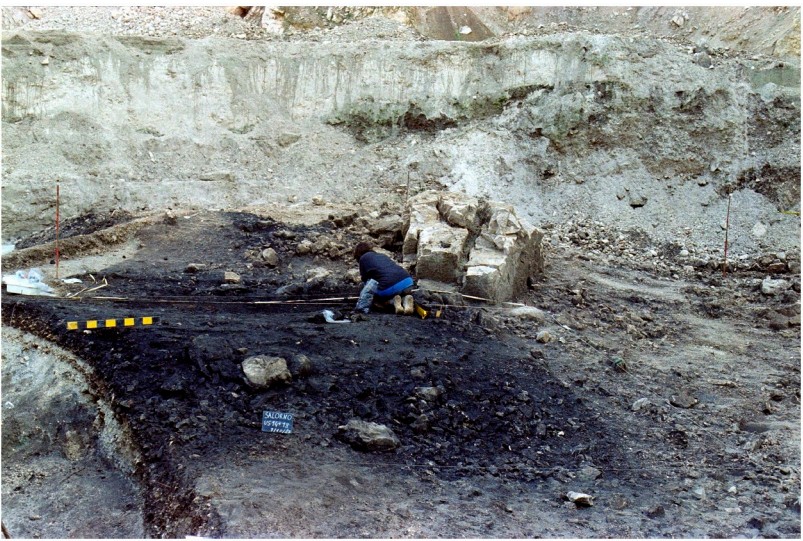

**Fig 2. The cremation platform during the excavation of 1987 (courtesy of Ufficio Beni Archeologici di Bolzano).**

On top of US 11, feature US 10 appeared to be the last anthropogenic level of the site. Its sloping was completely different from lower levels, with a thickness of 5 to 8 cm and an area of about 150 m$^2$. US 10 was made of selected and compacted earthy and gravelly material, appearing as the result of an intentional obliteration of the *ustrinum*, which preserved the Bronze Age levels undisturbed.

## Materials and methods

### Excavation coding

The excavation site of Salorno covered a total surface of 26 m$^2$, forming a rectangular shape of 6.5 m x 4 m, oriented Southeast-Northwest. The area was subdivided into a grid of 25x25 cm squares (for a total of 416) identified with alphabetical letters from A to R on the short side, and numbers from 1 to 28 on the long side. The stratigraphic excavation distinguished a series of features (named US, "unità stratigrafica"): US 10 at the very top; US 11 subdivided in four cuts (I-IV); US 12 subdivided in two cuts (I, II); US 13; US 14 subdivided in two cuts (I, II); US 15; US 16 subdivided in two cuts (I, II); US 17; US 18 subdivided in three cuts (I-III); US 19 subdivided in two cuts (I, II); and US 20. US 11 corresponded to the major extension of the burnt deposits, whereas other US (US 13, US 14, US 17, US 18 and US 19) named features of the terrain or of anthropogenic nature that overlapped with each other on the Northwest margin of the excavation.

The archaeological and bioarchaeological findings were carefully classified during excavation, consistently labelled with the US/cut/grid coding, and contained in thick plastic bags after flotations in water to get rid of soil residues. All specimens are kept in the storage of the Ufficio Beni Archeologici di Bolzano, identified by this coding system for repository purposes.

All necessary permits were obtained from the Ufficio Beni Archeologici di Bolzano for the described study, which complied with all relevant regulations.

### Bioanthropological analysis

A funeral pyre is by definition a place of "waste" elements, accumulated during and after the cremation of one or more bodies, and it is therefore an area of extreme fragmentation and indeterminacy. However, depending on the degree of preservation, deformation and fragmentation of the skeletal elements, human cremains can still provide important data for a bioanthropological analysis. Preliminary to the bioanthropological study, the authors went through a careful screening of the content of the bags to separate bone fragments from soil residues, organic material (charcoals, animal bones), and admixed archaeological items (pottery, metal, glass, carved bone) (Fig 3).

The bioanthropological analysis of the burnt skeletal and tooth fragments followed traditional protocols, as to include as many quantitative variables and macroscopic observations as possible–anatomical identification and maximum size of identifiable skeletal elements, their weight and degree of combustion, classification of heat-induced fractures [28, 29]. Certain skeletal elements such as diaphyses of long bones, mandible corpus, parts of the temporal bones comprising the petrous process and the mastoid, thick cranial bones at the parietal and occipital level, and unerupted teeth still included in the maxillary bones are all more resistant to destruction from combustion at high temperatures [3, 18, 30, 31]. The relative durability of these skeletal fragments is particularly useful for classical bioanthropological analyses such as determining the minimum number of individuals (MNI), assessing sex and age at death and recording potential pre- and post-cremation modifications. Comparative ethno-archaeological studies, along with forensic practice and modern crematoria, were consulted to hypothesise temperatures reached during cremation in open sites [32–36].

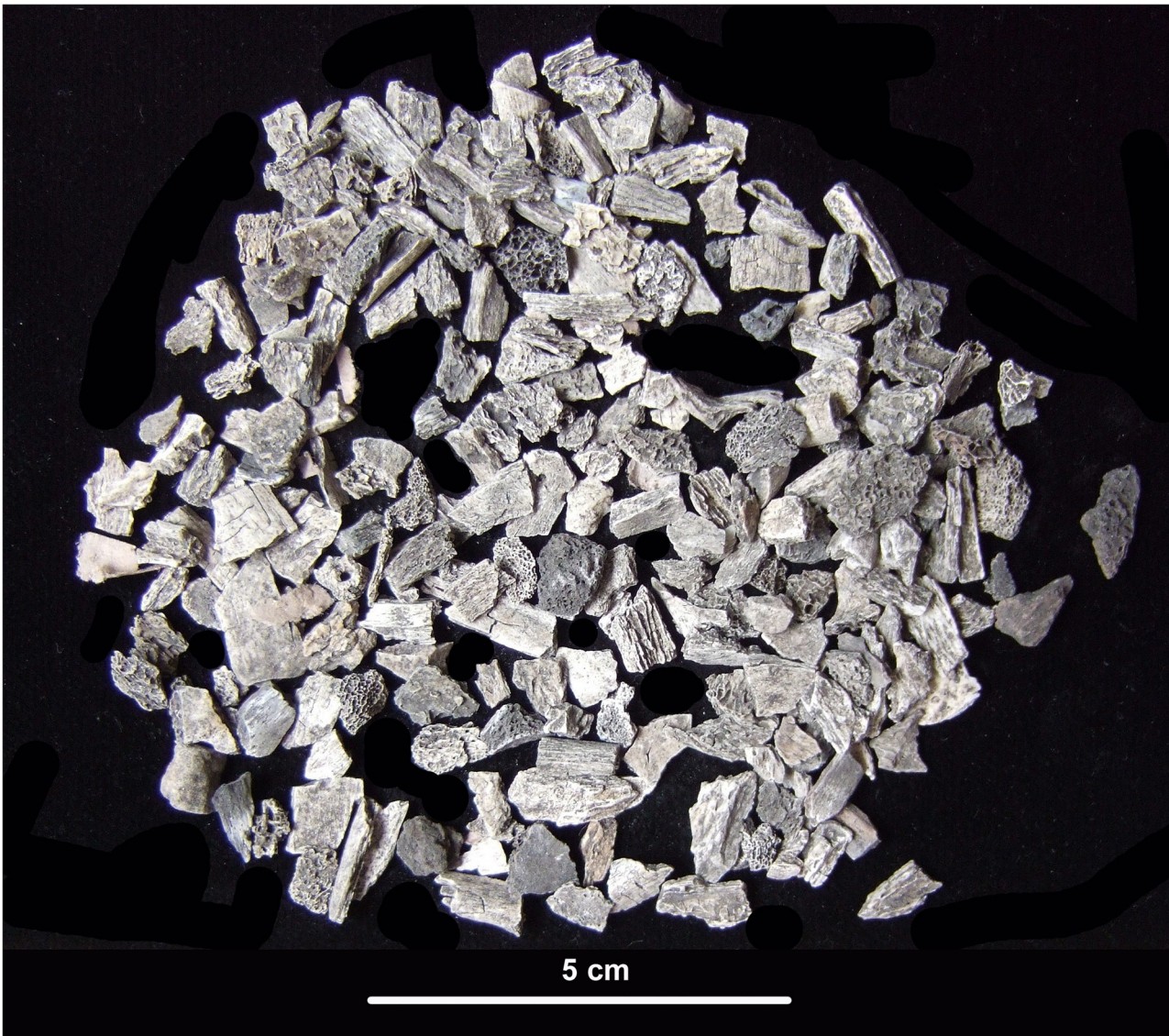

**Fig 3. Sample of cremated bones from feature 'US 11'.**

## Fragmentation analysis

In order to test the nature of the formation of such a quantity of human cremains in the sample of Salorno—Dos de la Forca, the authors decided to carefully examine the degree of fragmentation of the cremains along with their spatial distribution [37, 38]. Per each excavation square and level, the authors created sub-groups of bone fragments by size (> and < 20 mm), separating diagnostic and larger pieces from minute fragments. Where possible, fragments >20 mm were further subdivided by gross anatomical regions (cranial, post-cranial, teeth, undetermined).

The degree of fragmentation was assessed through a 'fragmentation index' calculated as the ratio between weights of the two subgroups. The higher the value, the higher the level of fragmentation.

'Fragmentation index' formula:

$$[(\text{weight of fragments} < 20\,\text{mm})/(\text{weight of fragments} > 20\,\text{mm})] * 100$$

A measure of the relative representation of the cranial and post-cranial bones was named *CPC Index* ('cranial/post-cranial index').

'CPC index' formula:

$$[(\text{weight of cranial bone fragments}/\text{weight of post} - \text{cranial bone fragments})] * 100$$

Whereas the 'fragmentation index' provides for a measure of the overall fragmentation of the sample, the 'CPC index' can provide a quantitative assessment for ascertaining post-cremation practices of selection and displacement of the cremains [38].

All raw data generated in the study are available in the S1 Table.

## Results

### Bioanthropological analysis

The human cremains from Salorno—Dos de la Forca show typical heat-induced bone alterations and fractures [39, 40]: longitudinal, transverse or reticular cracks, smoothing of the bone surface before splintering, U-shaped (or thumbnail) cracks typical of heat response on the diaphyses of long bones (Fig 4), and concentric splits of spongy bones. Crowns of erupted teeth are not preserved, while dental roots fragments are found with exposed dentine (Fig 5).

The vast majority of the human cremains are white-calcinated, suggesting that temperatures normally reached and exceeded 700˚C [41–46] causing complete dehydration of the bones. In more limited cases, bone fragments show blueish grey colours, while dark-brown or black charred chromatism are extremely rare.

The skeletal elements anatomically identified are all > 20 mm. Anatomical attribution of the bone fragments was not carried out for fragments < than 20 mm–with the exception of teeth and few clearly identifiable cranial fragments.

A total of 2,317 g of cranial bones and of 12,007 g of postcranial bones was identified. The total weight of diagnostic bones and teeth amounts to 635 g–about 1% of the total fragments' weight–of which 105 g of cranial and postcranial elements, and 530 g of tooth fragments. The minimum number of individuals (MNI) calculated with conventional procedures would be of four individuals: two adults, identified by a minimum number of two right mastoid processes and two right mandibular condyles; one child of about 6 years, identified by one right deciduous second molar and one left upper canine germ, both attributable to the same developmental stage (ca. 6 years ± 24 months) [47, 48]; and one juvenile individual of 12–14 years of age, identified by one right subadult head of radius. Obviously, the MNI thus calculated is evidently under-represented when compared with the total weight of human cremains found at the site.

The lack of clearly diagnostic sex indicators along with the high degree of fragmentation make the assessment of biological sex of single elements impossible.

### Fragmentation analysis

A total of 63,555 g of human cremains was identified from the burnt deposits of Salorno. Of these, 76.6% is represented by fragments <20 mm, 22.5% by fragments >20mm, and 0.8% by tooth fragments. The average fragmentation index is of 340 –meaning that in the Salorno sample minute bone fragments are over three times more frequent than larger ones.

In the group of fragments >20 mm, the cranial bone elements amount to 16.2% of the total weight, with a CPC index of 19.3. This amount is in line with expected figures calculated on

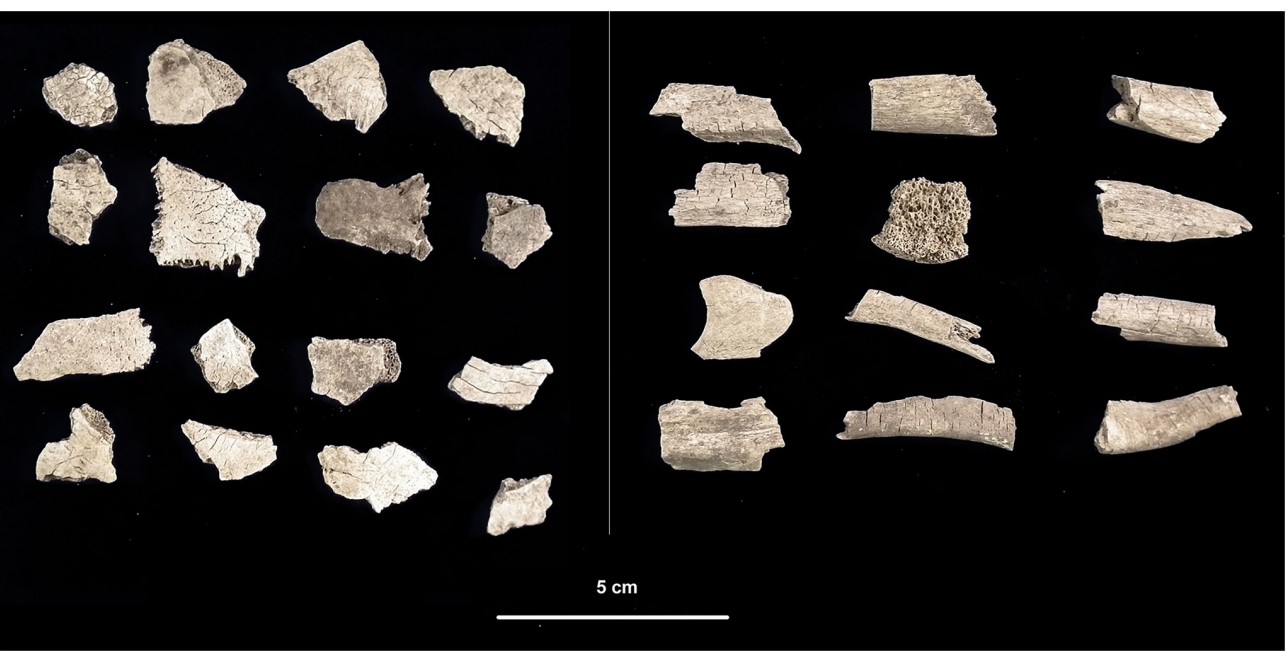

**Fig 4. Selection of cranial and long bones' diaphysis fragments from feature 'US 11', showing typical morphological and microstructural modifications due to high temperatures during cremation.**

the average variability of a human skeleton's weight and proportional weight of anatomical regions, calculated on complete, non-burnt adult skeletons [49].

## Spatial analysis

The detailed archaeological excavation documentation enabled the creation of a spatial model of intersecting squares for the whole excavation area. Burnt human bone remains are concentrated in feature US 11 (97.3% of the total, for a total of 61,832.1 g)–comprehensive of four horizontal parallel cuts, plus a surface layer named "roof". The other excavation features (US 12–19) provided only small scatters of bone fragments–with less than 2 kg in total. The authors have therefore performed the spatial distribution analysis of the bones only for US 11, considered largely representative of the whole sample. However, a complete database of the weights of the human skeletal remains is reported in S1 Table.

The burnt human bones from Salorno are concentrated in a relatively small (i.e. 22.8% of the excavation grid) central area of the excavation. Fig 6 illustrates the plan, divided by squares of 25x25 cm, and coloured following a proportional weight scale (see legend) to highlight areas with higher concentrations of human skeletal remains. Besides the main dispersal area, there are isolated accumulations of bones to the South [squares A26, B27, D25, E26; weight 403.2 g; average fragmentation index 136] and to the Northwest [squares C19, D19, G17, M15; weight 229 g; average fragmentation index 367]. Within the main concentration of the human cremains, the central squares in correspondence to squares of rows F/G/H-3/4/5 are those recording highest quantities of bone fragments (20,037 g). G5 provides for the highest cumulative amount of human cremains with 4,092 g, followed by G4 with 3,490.2 g. As shown in Fig 7, the density of bones decreases progressively by moving towards the perimeter of the area.

The distribution of the cranial and post-cranial bone fragments shows slight but interesting differences (Fig 8). While both categories of fragments are more concentrated in the centre of

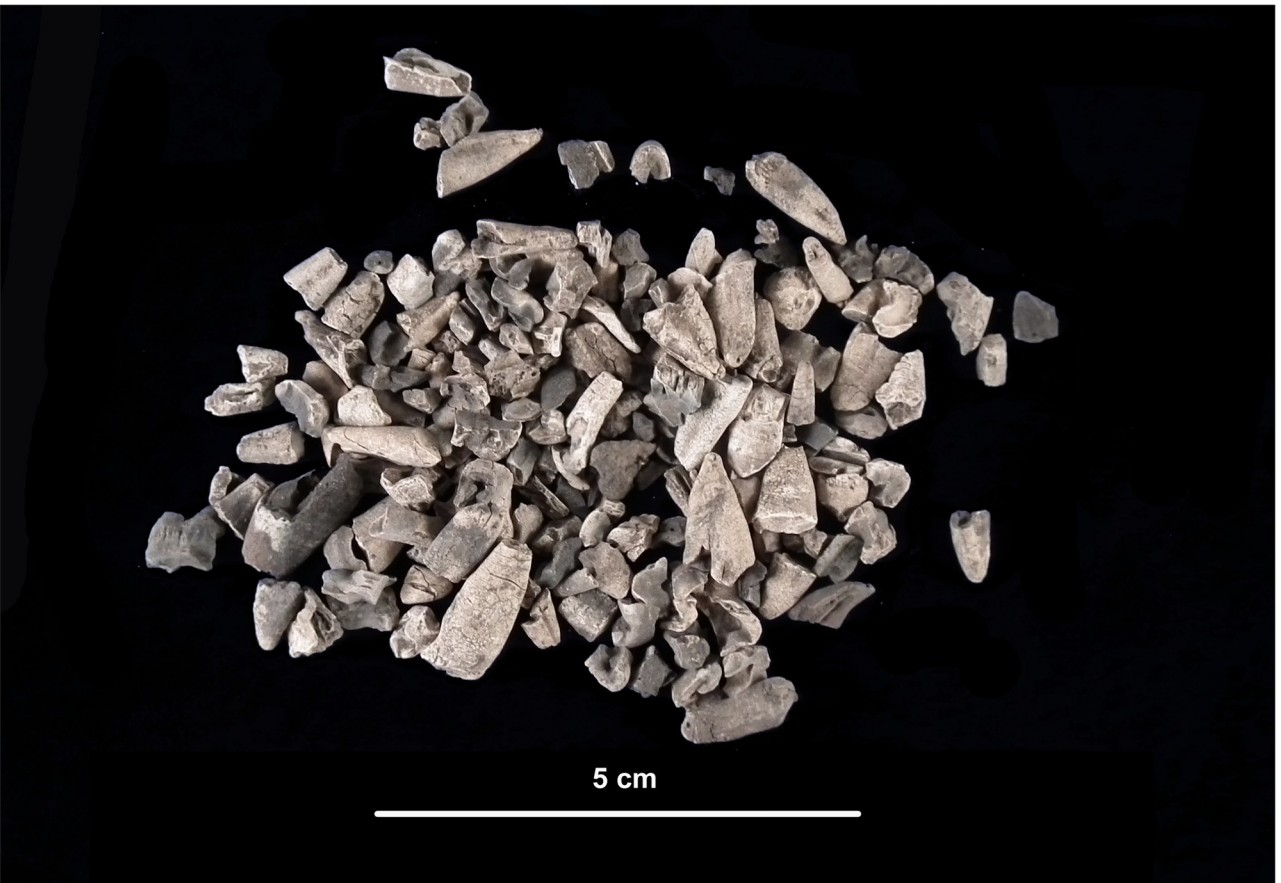

**Fig 5. Example of tooth fragments from feature "US 11".**

the *ustrinum* with an average CPC index of 16.2, cranial bones are less represented in the Northwest margin (e.g. squares of rows 8 and 9) where post-cranial bones are more common, as shown by a CPC index of 7.2.

## Discussion

The amount of cremated human remains found at the *ustrinum* of Salorno—Dos de la Forca (about 63.5 kg) is truly exceptional, by far the greatest when compared with known similar Italian contexts dated between the Late Bronze and the Iron Age, where only a handful of human bone remains are usually recovered [18]. Such data could depend on preservation bias, or rather show that the pyre area was used for few cremations at a time, with bone residuals being thoroughly wiped off after selection for internment. It is the case of the contemporary necropolis of Frattesina–Le Narde, in the Po Plains, where the *ustrinum* area is located among tombs and preserves little quantities of burnt remains–hardly any–including both pyre residuals and human bones. However, there are instances like the *ustrinum* of Vadena (South Tyrol) that show larger amounts of bone remains (i.e. 12 kg) [50].

The absence of a reference necropolis for Salorno cannot rule out simplistically that there was not one. For this reason, the authors have tested two hypotheses combining the results from the bioanthropological, spatial and fragmentation analyses. Supposing that Salorno may be the primary site of repeated cremations left *in situ* (hypothesis A), the expectation would be

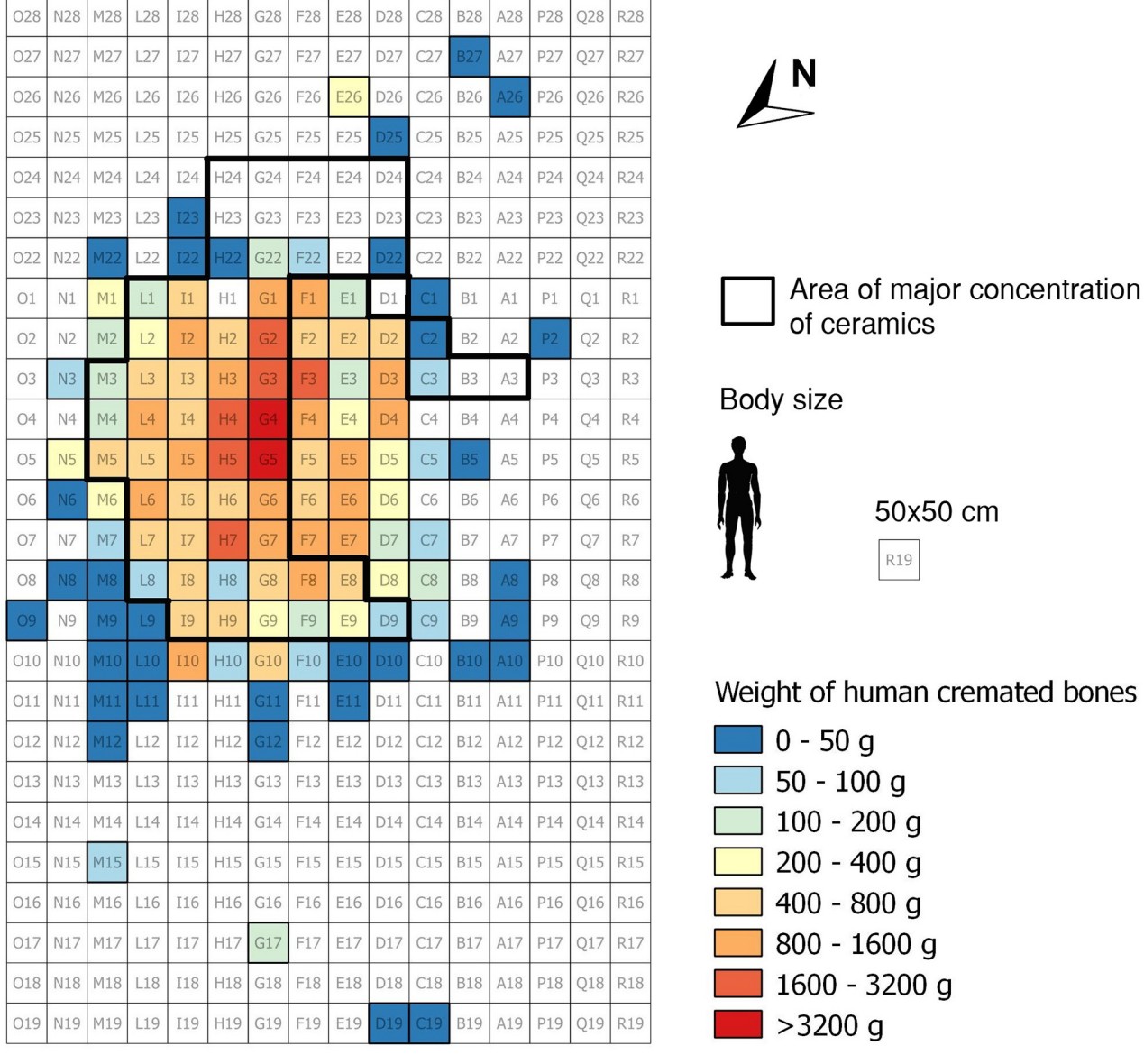

**Fig 6. Distribution of bone fragments on the excavated area of 25x25 cm squares.**

a coherent representation of all anatomical districts in the human cremains sample, with possibly more durable skeletal elements still identifiable. Vice versa, supposing that Salorno may be a traditional cremation site where the most intact skeletal elements were collected for secondary deposition in urns or pits (hypothesis B), the expectation would be to find less representative and more fragmented bone remains as residuals from the burning of the bodies.

The data from the osteo-dental analysis of Salorno confirmed the presence of both adult and sub-adult individuals, showing that the pyre was used with no distinction of age groups. In the absence of clear sex indicators from the few identifiable bone fragments, the commingled archaeological material can be of help in trying to identify if the pyre was utilized for both men and women. Published data on the pottery analysis found in the burnt deposits of Salorno shows a prevalence of fragments of cups (minimum number = 48), followed by truncated cone

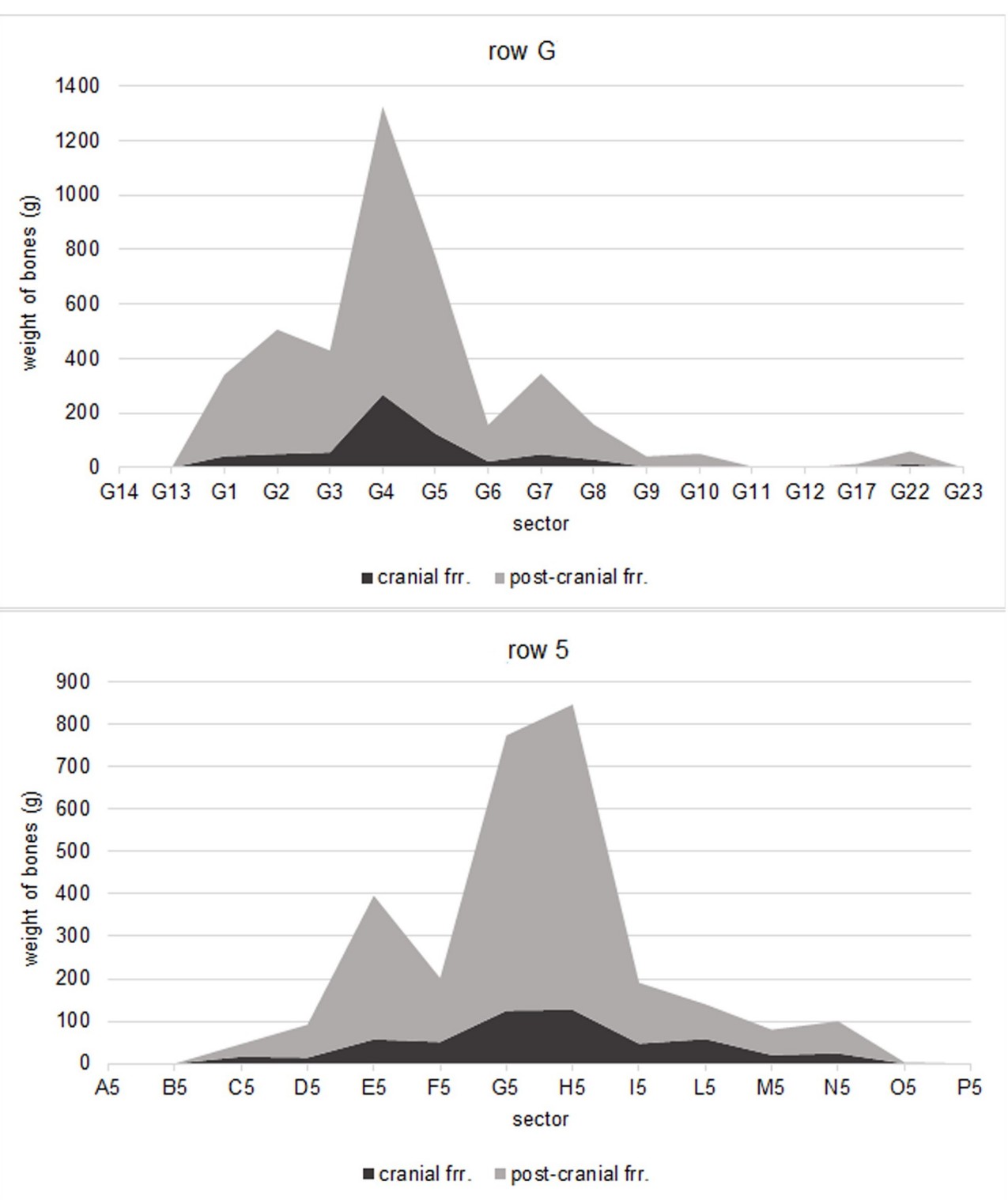

**Fig 7. Weights of bone fragments represented across the transversal sections of the excavated area of 25x25 cm squares: Transect along row G (top) and east-west transect along row 5 (bottom).**

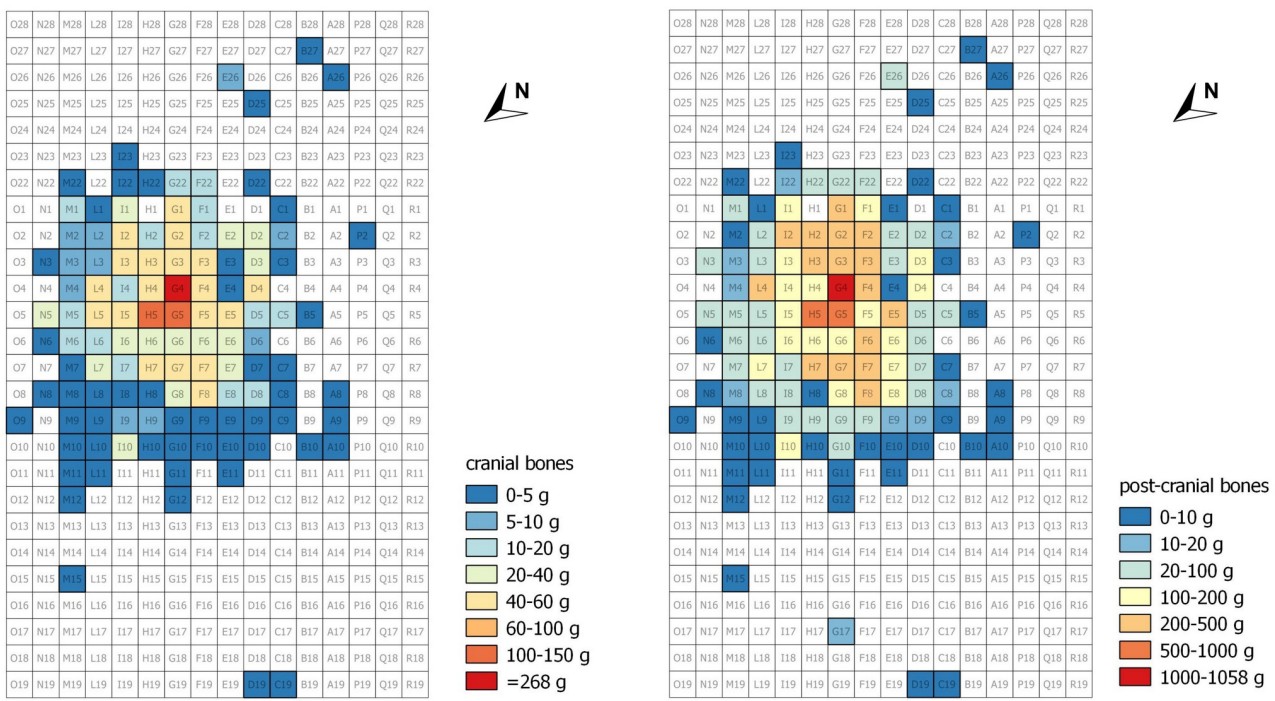

**Fig 8. Distribution of cranial (left) and post-cranial (right) fragments in the Salorno—Dos de la Forca *ustrinum* area.**

bowls (minimum number = 2 or 3), bowls (minimum number = 4), biconical vessels (minimum number = 5), and jars (minimum number = 2) [25]. All the pottery is typologically attributable to the Luco A *facies* (culture), datable to the Late Bronze Age (1150–950 BCE). Other archaeological findings include 6 spindle-whorls, along with a few glass beads (necklace elements) and bone discs decorated with circles–all interpreted as objects accompanying the deceased on the pyre (so-called 'pyre goods') (S4 Fig). Comparative Bronze Age burials show that adult males were normally buried without grave goods in these phases, making the absence of male grave goods at Salorno less significant [51, 52]. Concerning the presence of female individuals, the Final Bronze Age urnfield of Frattesina–Le Narde in the delta of the Po River, which counts about 800 burials, provides for 10 urns containing spindle whorls, along with human cremated bones and other goods. The osteological analysis [51, 52] confirms that these burials all belong to female or sub-adult individuals–these latter probably also female. Adult female grave goods are usually represented by a spindle-whorl and by one or more fibulae/rods, while graves of children or young women contain more items and eye-catching ornaments. Male tombs in Frattesina are slightly under-represented, with only two tombs containing fragments of two swords clearly associated with male grave goods.

As such, and with all due caution considering the fragmentation of the sample, the bioanthropological and archaeological analyses would point to Salorno as a site of primary cremation and deposition of the bodies, in support of hypothesis A. As a further element of analysis, the authors have modelled two scenarios to try to reconstruct the minimum number of individuals (MNI) here preserved.

## Simulating a minimum number of individuals

The paucity of diagnostic skeletal and dental elements preserved for a traditional bioanthropological analysis suggested a MNI of four individuals–two adults and two sub-adults of about 6

and 13 years. However, the amount of cremains along with the archaeological findings would suggest a considerably higher number of individuals. We have attempted some simulations through comparison with modern and ancient cremations.

Based on modern cremations, the weight of a cremated adult human skeleton can vary between about 2–3 kg for males, and about 1.5–2.5 kg for females [8, 53–55]. Pre-protohistoric human populations in Europe were characterised by lower body-mass than today [e.g. 56, 57], therefore we may use maximum weights of Bronze Age cremated bone depositions from Northern Italy [18] for more realistic figures: a burnt adult male skeleton would weight 2,500 g; an adult female 1,800 g; the skeleton of an infant or adolescent would be lighter in weight—about 500 g for the infant, by average of known weights, and 1,000 g for a young adolescent.

If we hypothesise that the cremation site of Salorno—Dos de la Forca was used by a community or social group with demographic characteristics typical of a pre-industrial population, and without any substantial demographic variation in time, we may assume that an equal number of adult (half males and the other half females) and sub-adult individuals were cremated. However, from studies on traditional, pre-industrial populations with no use of vaccines, we know that the mortality of sub-adults was around 50% of the total number of individuals [58–62].

Bones collected for deposition in urns in the necropolis Frattesina–Le Narde corresponds on average to about 40% of the expected weights but varies considerably for age class [51, 52]. By contrast, the only two cases documented for the final stages of the Bronze Age in the region are those of Collalbo and Laion–Novale di Sotto (Bozen) [63]. In each of the two sites, one single urn was found, belonging to an adult female (1,443 g) and an adult male (1,695 g) individual, respectively. These urns yielded quite high weights of cremated human remains–close to expected estimates of a complete *ossilegium*, with thorough collection of bone remains after the pyre. In the absence of a reference necropolis, we will use the above comparative data to model how many individuals could be represented at Salorno.

If we assume that we are dealing with a site of primary cremations, with cremated bones left *in situ* (hypothesis A), the total weight of the human cremains from Salorno (63,555 g) would suggest a minimum number of 48 individuals there cremated:

12 adult males * 2,500 g each +

12 adult females * 1,800 g each +

24 sub-adults * 500 g =

63,600 g.

If on the contrary we assume that Salorno is an accumulation of bone residuals after collection for deposition in burials (hypothesis B), we can use for reference the contemporary burials of Novale di Sotto for adult males (1,695 g), of Collalbo for adult females (1,443 g), along with the average weight of sub-adult burials from Frattesina–Le Narde (157 grams). By subtracting these values from expected weights of burnt skeletons, we obtain a minimum number of 172 individuals:

43 adult males * (2,500 g − 1,695 g) +

43 adult females * (1,800 g − 1,443 g) +

86 sub-adults + (500 g − 157 g) =

63,468 g.

These two simulations show that the minimum number of cremated individuals varies considerably depending on the assumptions. The analysis of the spatial distribution of the human cremains from Salorno can help shed further light on this matter.

## Inferences from the spatial distribution of the human cremains

The spatial analysis of the human bones' accumulation and distribution at Salorno would point to a preferred location for the set-up of the pyre/pyres, around squares G4 and G5. The area of major density of bone fragments (delimited by rows 1–11 and columns L-D, and reaching maximum densities in squares F/G/H-3/4/5 with average weight of 2,226.3 g) measures 2.5 m by 1.5 m. This seems a realistic measure for a pyre that could have hosted at most two bodies side by side at the same time–although the practice of burning multiple bodies stacked together on top of each other cannot be excluded. This would suggest a preferable, albeit not rigidly fixed, orientation of the corpses on the pyre along the Northwest–Southeast axis and it would further support hypothesis A, i.e. that the burnt remains of the deceased were left *in situ*.

As seen in the result section, cranial fragments from the burnt deposits of Salorno amount to 16% of the total weight. Cranial bones represent on average about 15% of the total weight of a human skeleton–calculated as a mean value across different developmental stages [64–67]. Furthermore, published studies on the ratio between weights of cranial and post-cranial fragments in protohistoric funerary contexts show a relative selective representation of the cranial skeleton [22, 37], suggesting an intentional and preferential selection of the head/face remains of the deceased for the burial. Both the identification of a preferential spot and orientation of the body on the pyre, along with expected representations of the cranial and post-cranial skeletons in the cremation site of Salorno would both support hypothesis A.

Although hypothesis A seems to be confirmed by the bioanthropological, archaeological and spatial analyses, the possibility that Salorno could be the result of a series of cremations with selection of bones for redeposition in burials cannot be ruled out completely. Indeed, the expectation in this case would be the finding of an extremely fragmented sample of human cremains, with hardly any diagnostic bone element, which is exactly the case. However, the extreme degree of fragmentation of the human cremains from Salorno, besides being interpreted as proof of anthropogenic factors that would point to a repeated use of the pyre with only residuals from collections staying *in situ*, could also be ascribed to natural physical causes (combustion temperature; cyclical seasoning and thawing of the bone in case of reasonable freezing temperature during Alpine winters), and even to post-depositional factors (the material was admixed with gravels that could have caused further bone fragmentation by friction).

## Cremation rituals at Salorno: Some hypotheses

If, as discussed thus far, this is both the primary cremation site and the burial site of a small social group of the Late Bronze Age in the Italian Alpine region, Salorno—Dos de la Forca becomes a unique case study to infer aspects of an ancient cremation ritual in its complexity. Whereas cremains from contemporary cemeteries are essential for a better understanding of the post-cremation phases, they lack information about the preliminary and central phases of the cremation ritual.

For example, would it be possible to determine whether at Salorno bodies were cremated before or after decomposition? Published studies address the correlation between warping and thumbnails fractures and bone alterations due to cremations operated on fleshed bodies [42, 68–73], stating that it is difficult to establish with certainty whether bodies were cremated soon after death or after the decomposition of soft tissues. In Northern Italy there is no evidence of contemporary ossuaries–natural or artificial structures for housing corpses for the

decomposition of skeletons. Furthermore, the presence of burnt personal ornaments, which were part of the dead's clothes are found at Salorno. We can therefore assume that bodies were here cremated before the decomposition of soft tissues.

The prevalence of cups (at least 48) among the pottery found at Salorno is also of particular interest in the reconstruction of the ritual. As Renato Perini points out [74], cups are often what prevail in places of worship of the Luco area–for example, in South Tyrol at Burgstall, Plörg, and Seeberg, and in Trentino at Montesei di Serso [74–78]. Luco A type is interpreted as tableware for drinks and solid foods, which were probably consumed during the funerary rituals on site and then thrown into the pyre (Fig 9).

There is a large overlap in the distribution of bone and pottery fragments at Salorno (Fig 6). The pottery shards form a sort of semicircle around the highest concentration of cremated human bones. In the Southeast part of the pyre area, corresponding to the two rows of squares H24-D24 and H23-D23, pottery fragments are the only findings, with no human remains admixed. Such a distribution of the pottery could further support the hypothesis that the head of the dead was oriented to Southeast. By standing next to the deceased's body, the participants to the ritual could be safely protected from the fire while making offerings, by throwing their cups in the pyre as part of the libation ritual.

Interestingly, the minimum number of Luco-type cups found admixed with cremated bones at Salorno points to a striking coincidence with the MNI of 48 individuals suggested for validating hypothesis A. The pottery analysis restricts the chronology of Salorno over a time span of about 200 years, roughly corresponding to 8 generations of 25 years each. Assuming that each deceased was offered one cup during the cremation ritual, this number could point to a small community of a single line of descent over 8 generations–e.g. 8 succeeding nuclear families of 6 people each.

The Luco type pottery vessels found in the burnt deposits of Salorno are of further relevance for discussion about the practice of the *Brandopferplätze*, "votive pyres" common in the Alpine areas between the Bronze and Iron Age.

**Salorno—Dos de la Forca in the context of the *Brandopferplätze*.** *Brandopferplätze* are ceremonial sites normally located on mountain tops or across borderlands, typified by animal sacrifice, along with the practice of deliberate fragmentation of vessels which are left *in situ*. In particular, the frequency of Luco type jugs and cups is higher in ceremonial sites than in contemporary settlements, showing a preference for specific ceramic types for ritual practices [77, 79, 80].

Similarities with Salorno may seem clear, except that *Brandopferplätze* are never associated with either human bones, or grave goods. However, some shared aspects are unquestionable and may support the identification of Salorno as a hybrid ceremonial site: first of all, fire is the means for rituals of destruction–not only of the deceased, but also of animal and other organic plant offerings, along with pottery; secondly, ceramic containers like jugs and cups are broken and left behind, presumably after use for ceremonial libations and therefore impregnated of symbolic value; thirdly, the ceremonial site coincides with a territorial landmark, shared across bordering communities; lastly, the site is repeatedly used over a number of generations (e.g. Sciliar and Seeberg [77]).

Known archaeological references from the South Tyrol and Trentino show that houses in contemporary settlements of the Luco A area, such as Appiano [81], Ganglegg [78] and Montesei di Serso [75], hosted one or a maximum of two nuclear families of 6 to 12 individuals. For comparison, the individuals cremated at Salorno could belong to social groups of this kind, perpetuated for a maximum of 8 generations. Judging from the prestige of the goods found associated with the bones, it is also plausible that the group was part of local elites.

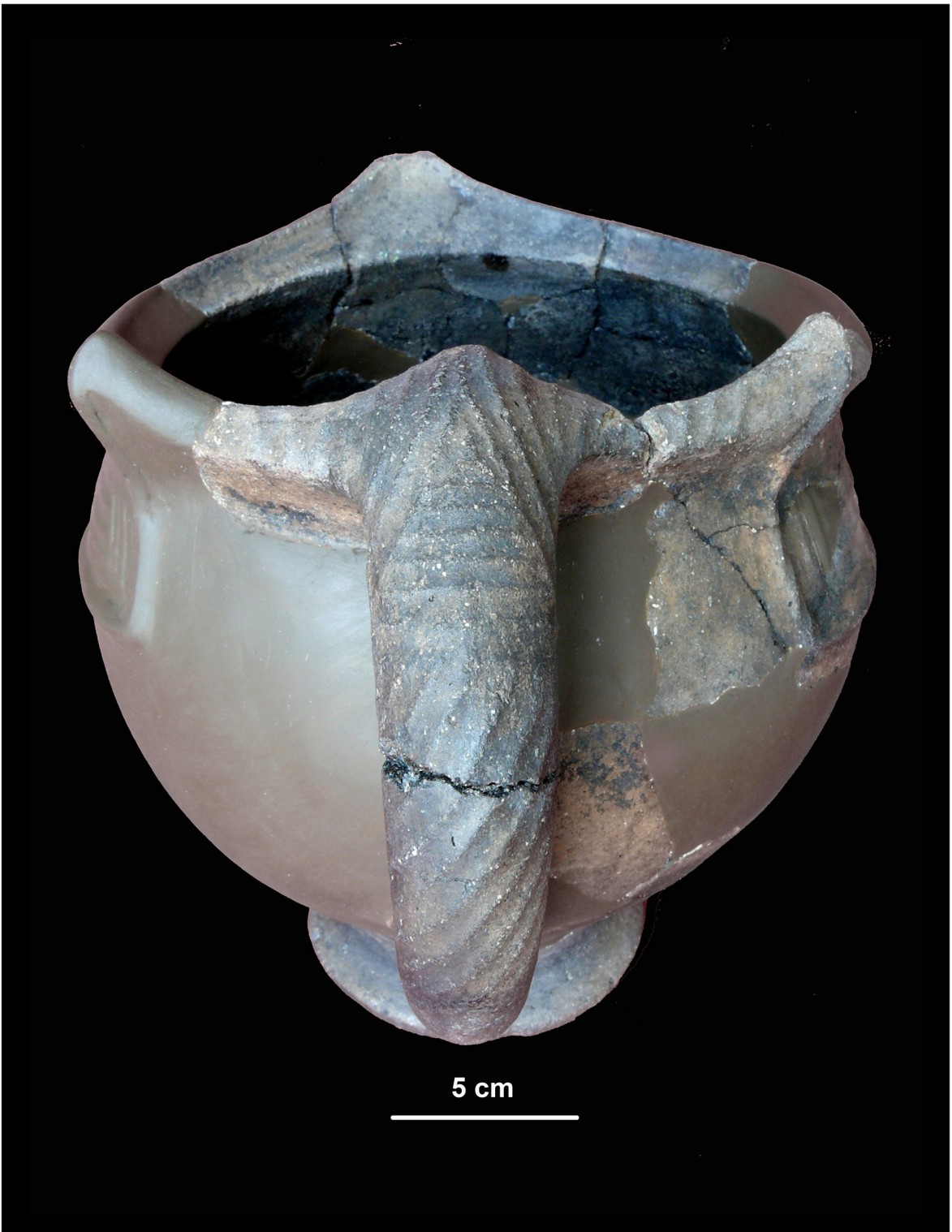

**Fig 9. Luco A type vessel (photo by Günther Niederwanger, courtesy of Ufficio Beni Archeologici di Bolzano).**

## Conclusions

The site of Salorno—Dos de la Forca described in this study is a truly extraordinary find in the context of the Late Bronze Age funerary archaeology of Northern Italy. Not only does the quantity of human cremains exceed that of any other contemporary or subsequent contexts interpreted as *ustrina*, but this study would even frame it as a new typological and functional category within the Bronze Age ceremonial sites of Northern Italy. Indeed, Salorno cannot be defined as a *ustrinum* in the strict sense: high quantities of human bones left in place would address it as a place for both cremation and final deposition of the deceased. Salorno appears as different both from the typical large 'urnfields' documented in the Po area, and from the small burial grounds/isolated graves known from adjacent sites in the Alpine valleys. Here, the authors interpret Salorno as the product of a complex series of rituals in which the cremated remains of the deceased did not receive individual burial, but were left *in situ*–in a collective/communal place of primary combustion, defining an area of repeated funeral ceremonies involving offerings and libations across a few generations. In this context, similarities with the Alpine cult places called *Brandopferplätze* may suggest a sort of syncretism between funerary and other ritual practices.

With Salorno—Dos de la Forca, the variability of mortuary customs at the end of the Bronze Age in the Alpine area is further populated, being added to known evidence of small burial grounds with cinerary urns, isolated urns, and human remains deposed in various areas of the settlements. At a time of strong 'globalizing' trends generated by the widespread demand for Alpine copper [82], a more private ritual could reflect a push for asserting one community's own identity.

Ancient-DNA studies suggest great mobility and interconnection among European communities of the Bronze Age [83–87]. Being cremations the dominant form of burial in this historical period, with virtually no chance of preservation of collagen, other techniques may contribute to explore the permeability of society to external influences, such as strontium isotope analysis [88–92]. Future interdisciplinary investigations of the material found at Salorno could shed light on local histories that would otherwise remain without memory.

## Supporting information

**S1 Fig. The site of Salorno—Dos de la Forca in the context of the Adige river valley (courtesy of Ufficio Beni Archeologici di Bolzano).**
(TIF)

**S2 Fig. Selection of materials from feature 'US11' (top: glass beads forming a necklace; bottom: antler discs and gold hair-ring) (courtesy of Ufficio Beni Archeologici di Bolzano).**
(TIF)

**S3 Fig. Concentration of fragmented pottery from feature 'US11' (courtesy of Ufficio Beni Archeologici di Bolzano).**
(TIF)

**S4 Fig. Spindle-whorl from Salorno (courtesy of Ufficio Beni Archeologici di Bolzano).**
(TIF)

**S1 Table. Table of weights of the dental and skeletal fragments per excavation square.**
(XLSX)

## Acknowledgments

The authors are indebted with the Ufficio Beni Archeologici di Bolzano previous and current directors, Dr Lorenzo Dal Ri and Dr Catrin Marzoli, who have greatly supported and encouraged this scientific contribution. Special thanks are directed to the staff of Ufficio Beni Archeologici di Bolzano, in particular to Mr Roland Messner and Mr Gianni Santuari. Important unpublished information about the excavation of the site were kindly provided through discussions with Dr Lorenzo Dal Ri and Mr Gianni Rizzi who personally conducted often complicated archaeological excavations for Società Ricerche Archeologiche Rizzi snc (Bressanone). The authors would also like to thank Mr Luca Pisoni for useful discussions and interpretations of the spatial analysis of the site. We thank Giovanna Fregni for a final revision and proofreading of the manuscript. Finally, the authors are indebted to Dr Lynne A Schepartz and another anonymous reviewer for their insightful suggestions that have highly improved the quality of the manuscript.

## Author Contributions

**Conceptualization:** Federica Crivellaro, Claudio Cavazzuti.

**Data curation:** Federica Crivellaro, Francesca Candilio, Umberto Tecchiati.

**Formal analysis:** Federica Crivellaro.

**Funding acquisition:** Umberto Tecchiati.

**Investigation:** Federica Crivellaro, Claudio Cavazzuti, Francesca Candilio, Umberto Tecchiati.

**Methodology:** Federica Crivellaro, Claudio Cavazzuti, Francesca Candilio, Alfredo Coppa, Umberto Tecchiati.

**Project administration:** Umberto Tecchiati.

**Supervision:** Federica Crivellaro, Alfredo Coppa, Umberto Tecchiati.

**Validation:** Federica Crivellaro, Claudio Cavazzuti, Umberto Tecchiati.

**Writing – original draft:** Federica Crivellaro, Claudio Cavazzuti.

**Writing – review & editing:** Federica Crivellaro, Claudio Cavazzuti, Umberto Tecchiati.

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
