## [Decision Letter · Decision Letter 0]

30 Dec 2021

PONE-D-21-34557Salorno – Dos de la Forca (Adige Valley, Northern Italy): a unique cremation site of the Late Bronze Age.PLOS ONE

Dear Dr. Tecchiati,

Thank you for submitting your manuscript to PLOS ONE. After careful consideration, we feel that it has merit but does not fully meet PLOS ONE’s publication criteria as it currently stands. Therefore, we invite you to submit a revised version of the manuscript that addresses the points raised during the review process.

We look forward to receiving your revised manuscript.

Kind regards,

Lynne A Schepartz

Academic Editor

PLOS ONE

Journal Requirements:

2. In your manuscript, please provide additional information regarding the specimens used in your study. Ensure that you have reported specimen numbers and complete repository information, including museum name and geographic location. 

3. We note that you have stated that you will provide repository information for your data at acceptance. Should your manuscript be accepted for publication, we will hold it until you provide the relevant accession numbers or DOIs necessary to access your data. If you wish to make changes to your Data Availability statement, please describe these changes in your cover letter and we will update your Data Availability statement to reflect the information you provide

Reviewers' comments:

Reviewer's Responses to Questions

**Comments to the Author**

1. Is the manuscript technically sound, and do the data support the conclusions?

Reviewer #1: Partly

Reviewer #2: Partly

2. Has the statistical analysis been performed appropriately and rigorously? 

Reviewer #1: N/A

Reviewer #2: No

3. Have the authors made all data underlying the findings in their manuscript fully available?

Reviewer #1: Yes

Reviewer #2: Yes

4. Is the manuscript presented in an intelligible fashion and written in standard English?

Reviewer #1: Yes

Reviewer #2: No

5. Review Comments to the Author

Reviewer #1: Cremation studies of archaeological material are to be commended and encouraged rather than ignored. This study aims to make sense of what appears to have been a large cremation platform or ustrinum with a deposit of burned bone and other artifacts from the Late Bronze Age in northern Italy. The authors assume that the bone was burned from fleshed bodies, but they need to state why this assumption is made. Could the bone have been burned dry? For example, there is no reference to warpage or shrinkage of the bone. Furthermore, the bone looks not to have been calcined which indicates that the funeral pyre may not have reached extreme temperatures? The authors could include another natural factor for the fragmentation of the bone that has been observed and that would be from cyclical seasonal freezing and thawing in an alpine environment (if water could have reached the bone). The authors use weight of the cremated remains to suggest MNI. The normal distribution of identifiable fragments from a pyre are at the head and feet in an extended body position with the fire centered at the center of the body, and with little identifiable from the center of the body. What needs to be clarified is why the greatest density of bone is in the center of the archaeological deposit and both cranial and post-cranial elements are represented in this area. Individuals of various ages at death are represented among the cremated remains. It was less clear, other than inferred via artifacts within the deposit, that males and females were represented. This should be made clear from the human skeletal remains if possible, and if not, it should be stated outright that biological sex indicators were not preserved. Results of DNA analyses on burned bone are making strides. I would not count out the possibility for DNA analyses to be undertaken on this material, but rather would ask what would be addressed by undertaking this possible line of research? Is there any inference from the archaeological evidence of the size of the associated settlement to compare with the hypothesized number of individuals represented? Are there inhumation burials from the same time period in the region or is cremation the predominant form of disposal of the dead at this time and place? Lastly, a minor suggestion would be to replace "physical anthropology" with "biological anthropology".

Reviewer #2: The subject of this manuscript is very interesting, and more work on human cremations is certainly needed. The authors have a good opportunity and the data to produce work that would be of interest to a range of skeletal biologists, forensic scientists, and Classical scholars.

In its current form, the manuscript requires extensive revisions, re-arranging of the content, and a major re-write as the English is not at a publishable level and is difficult to understand in several places. The authors must seek the services of a technical editor with familiarity with the subject matter.

Here is an example where the English is impeding the author’s presentation:

In the abstract ‘Despite extreme bone fragmentation and lack of information per single individual, the study could benefit from accurate spatial documentation during excavation.’

This should be re-written as ‘The patterning of bone fragmentation and commingling of individuals was investigated using spatial data recorded during the excavation’ to make it clear that the spatial analysis was undertaken as the data were indeed collected during excavation.

I provide the following comments to assist the authors, should they elect to make the revisions.

The authors do not present the problem they address effectively. The general issue is whether known characteristics of human cremains, from modern practices and protohistoric and contemporaneous archaeological deposits can help to understand the formation of the unusual burnt deposits at Salorno- Dos de la Forca. Were these evidence of repeated cremations left in situ, or the fragments remaining after select elements were removed for interment in urns at currently unknown tombs? The size, weight and spatial distribution of the cremains, along with the associated archaeological remains, are used to consider these two hypotheses. In the abstract the authors phrase the hypotheses in terms of their use in estimating the MNI, but this is only the most proximate use of them for understanding the deposits. In the body of the text, the hypotheses are not explicitly stated until p.12 in the results—and then again in the discussion. I recommend that the authors reorganize the manuscript to place the research question and hypotheses just before the Materials and Methods.

Materials and Methods: The authors should tailor this section (with subheadings and better organization) to specifically reflect how the materials and the methodology are designed to evaluate those hypotheses. Some results of the fragmentation analyses are included in this section and should be moved to the results (along with the ref to Table S1).

The fragmentation analysis can be better presented and some references on the relative durability of certain skeletal components (femoral diaphysis, mandible corpus, etc. should be included here instead of the comment on selective representation of the cranium. The use of the C index is fine, but calling it a ‘cranial index’ is not appropriate as a specific cranial index already exists. Is it not a cranial/postcranial index?

The last paragraph of the M & M on p. 10 needs to be included with the results.

Archaeological context: I have difficulty with the terminology in this section. The layer of major concern is US11, but the authors refer to US14 and 18 as concentrations of sherds within that layer. Are these then features, and not layers? Some of the information in this section is not concise or belongs in the discussion as interpretation of the behaviors leading to the deposits is offered. The important information is the spatial definition of the pyre area and its temporal limits. I get the impression that the authors could use the spatial information more effectively for the analysis of Salorno as the comparison with other sites to give a better impression of the site size and density that may reflect on its formation.

Results: Needs subheadings. Begin with the spatial analysis results on p. 12 and the paragraph mentioned above in the M & M. Much of this section is interpretation and not results. Separate out the results and discussion the interpretation in the discussion. This includes the comparisons with other sites, unless you did a specific analysis of the samples.

Remove the two hypotheses section from p. 12.

Identification and fragmentation

Shouldn’t the results of the two indices be here? Not in the discussion, where you contextualize them relative to other sites???

P. 11 line 255-256 seems to confuse use of maxillary for the upper jaw and mandible for the lower. The last line (261-262) of this paragraph makes no sense.

Given what can be identified, which is what % of the total fragment weight, what would be the MNI and age and sex distribution using conventional procedures?

Some interpretation of the fragmentation analysis is given in the Results, but this should be in the discussion.

Discussion: Reorganize to include information moved from earlier sections. The table with the other sites is not needed and can be stated in the text. Be clear about why estimates for males and females are from different sites.

The last section of the discussion, beginning with line 436, is good.

References: need to be standardized to follow PLOS ONE formatting; Very inconsistent.

Images:

-Is it possible to provide a map that includes the location of Salorno and the other sites used for comparison?

-The view of the valley is not very informative. Delete

-The images of bone condition and fragmentation, in situ and after sorting, are good; the images of the artifacts are less pertinent to the analysis presented here and most can be omitted. An image of the ritual cups is good as those are part of the material considered.

6. PLOS authors have the option to publish the peer review history of their article (what does this mean?). If published, this will include your full peer review and any attached files.

Reviewer #1: No

Reviewer #2: **Yes: **Lynne A Schepartz

---

## [Author Response · Author response to Decision Letter 0]

18 Mar 2022

Response to reviewers for PlosONE revision PONE-D-21-34557

We are very thankful for the reviewers’ insightful and useful suggestions that we believe have highly improved the quality of our manuscript. We report below, in red ink, detailed responses to the reviewers’ points.

REV. 1

• Fragmentation: The authors assume that the bone was burned from fleshed bodies, but they need to state why this assumption is made. Could the bone have been burned dry? For example, there is no reference to warpage or shrinkage of the bone. Furthermore, the bone looks not to have been calcined which indicates that the funeral pyre may not have reached extreme temperatures?

We have added a better description of the typology of bone alterations associated with the effects of extreme temperatures in the Results section: “The human cremains from Salorno – Dos de la Forca show typical heat-induced bone alterations and fractures [40,41]: longitudinal, transverse or reticular cracks, smoothing of the bone surface before splintering, U-shaped (or thumbnail) cracks typical of heat response on the diaphyses of long bones (Fig 4), and concentric splits of spongy bones. Crowns of erupted teeth are not preserved, while dental roots fragments are found with exposed dentine (Fig 5). 

The vast majority of the human cremains are white-calcinated, suggesting that temperatures normally reached and exceeded 700°C [42–47] causing complete dehydration of the bones. In more limited cases, bone fragments show blueish grey colours, while dark-brown or black charred chromatism are extremely rare”. 

Also, we have discussed the possibility that the bones could have been burned dry in the Discussion section ‘Cremation rituals at Salorno: some hypotheses’: “For example, would it be possible to determine whether at Salorno bodies were cremated before or after decomposition? Published studies address the correlation between warping and thumbnails fractures and bone alterations due to cremations operated on fleshed bodies [42,71–76], stating that it is difficult to establish with certainty whether bodies were cremated soon after death or after the decomposition of soft tissues. In Northern Italy there is no evidence of contemporary ossuaries – natural or artificial structures for housing corpses for the decomposition of skeletons. Furthermore, the presence of burnt personal ornaments, which were part of the dead’s clothes are found at Salorno. We can therefore assume that bodies were here cremated before the decomposition of soft tissues”.

• Fragmentation: The authors could include another natural factor for the fragmentation of the bone that has been observed and that would be from cyclical seasonal freezing and thawing in an alpine environment (if water could have reached the bone).

We have included this suggestion in the Discussion section: "[…] could also be ascribed to natural physical causes (combustion temperature; cyclical seasoning and thawing of the bone in case of reasonable freezing temperature during Alpine winters), and even to post-depositional factors (the material was admixed with gravels that could have caused further bone fragmentation by friction)”. 

• Position of anatomical districts: What needs to be clarified is why the greatest density of bone is in the center of the archaeological deposit and both cranial and post-cranial elements are represented in this area.

Thanks for the comment. The position of the pyre was probably not fixed as shown by the commingling of cranial and post-cranial fragments. We have better clarified this point in the Discussion section: “The spatial analysis of the human bones’ accumulation and distribution would point to a preferred location for the set-up of the pyre/pyres, around squares G4 and G5. The area of major density of bone fragments (delimited by rows 1-11 and columns L-D, and reaching maximum densities in squares F/G/H-3/4/5 with average weight of 2,226.3 g) measures 2.5 m by 1.5 m. This seems a realistic measure for a pyre that could have hosted at most two bodies side by side at the same time – although the practice of burning multiple bodies stacked together on top of each other cannot be excluded. This would suggest a preferable, albeit not rigidly fixed, orientation of the corpses on the pyre along the Northwest – Southeast axis and it would further support hypothesis A, i.e. that the burnt remains of the deceased were left in situ”.

• Sex: It was less clear, other than inferred via artifacts within the deposit, that males and females were represented. This should be made clear from the human skeletal remains if possible, and if not, it should be stated outright that biological sex indicators were not preserved.

We have clarified in the Results section that sex indicators were not available for a biological sex determination of the sample: “The lack of clearly diagnostic sex indicators along with the high degree of fragmentation make the assessment of biological sex of single elements impossible”. Also, in the Discussion section, we substantiate the validity of accompanying archaeological goods as proxies for sex determination from comparison with coeval regional contexts: “In the absence of clear sex indicators from the few identifiable bone fragments, the commingled archaeological material can be of help in trying to identify if the pyre was utilized for both men and women. Published data on the pottery analysis found in the burnt deposits of Salorno shows a prevalence of fragments of cups (minimum number = 48), followed by truncated cone bowls (minimum number = 2 or 3), bowls (minimum number = 4), biconical vessels (minimum number = 5), and jars (minimum number = 2) [26]. All the pottery is typologically attributable to the Luco A facies (culture), datable to the Late Bronze Age (1150-950 BCE). Other archaeological findings include 6 spindle-whorls, along with a few glass beads (necklace elements) and bone discs decorated with circles – all interpreted as objects accompanying the deceased on the pyre (so-called ‘pyre goods’) (S5 Fig). Comparative Bronze Age burials show that adult males were normally buried without grave goods in these phases, making the absence of male grave goods at Salorno less significant [52,53]. Concerning the presence of female individuals, the Final Bronze Age urnfield of Frattesina – Le Narde in the delta of the Po River, which counts about 800 burials, provides for 10 urns containing spindle whorls, along with human cremated bones and other goods. The osteological analysis [52,53] confirms that these burials all belong to female or sub-adult individuals – these latter probably also female. Adult female grave goods are usually represented by a spindle-whorl and by one or more fibulae/rods, while graves of children or young women contain more items and eye-catching ornaments. Male tombs in Frattesina are slightly under-represented, with only two tombs containing fragments of two swords clearly associated with male grave goods”. 

• DNA: Results of DNA analyses on burned bone are making strides. I would not count out the possibility for DNA analyses to be undertaken on this material.

We removed the sentence and rephrased in the Conclusions as follows: “Ancient-DNA studies suggest great mobility and interconnection among European communities of the Bronze Age [86–90]. Being cremations the dominant form of burial in this historical period, with virtually no chance of preservation of collagen, other techniques may contribute to explore the permeability of society to external influences, such as strontium isotope analysis [91–95]”.

• Settlement size: Is there any inference from the archaeological evidence of the size of the associated settlement to compare with the hypothesized number of individuals represented?

Unfortunately there are no published, coeval archaeological settlements excavated in their total extension for reliable demographic comparative analyses.

• Inhumations: Are there inhumation burials from the same time period in the region or is cremation the predominant form of disposal of the dead at this time and place?

Cremation is the predominant form of disposal of the dead at this time and place, and inhumations are extremely rare.

• Replace "physical anthropology" with "biological anthropology".

Thank you, we replaced it.

REV. 2

• State the problem more explicitly: The authors do not present the problem they address effectively. The general issue is whether known characteristics of human cremains, from modern practices and protohistoric and contemporaneous archaeological deposits can help to understand the formation of the unusual burnt deposits at Salorno-Dos de la Forca. Were these evidence of repeated cremations left in situ, or the fragments remaining after select elements were removed for interment in urns at currently unknown tombs? The size, weight and spatial distribution of the cremains, along with the associated archaeological remains, are used to consider these two hypotheses. In the abstract the authors phrase the hypotheses in terms of their use in estimating the MNI, but this is only the most proximate use of them for understanding the deposits. In the body of the text, the hypotheses are not explicitly stated until p.12 in the results—and then again in the discussion. I recommend that the authors reorganize the manuscript to place the research question and hypotheses just before the Materials and Methods.

Thank you for helping us with the restructuring of the research focus around these two main hypotheses. Indeed, we have followed the reviewer’s suggestion of clearly stating the two hypotheses in the Abstract (lines 37-39), placing them in the Introduction section (lines 71-73) and removing them from former page 12. The manuscript has been thoroughly reorganized in all its sections, adding subheadings, and moving and rearranging text that was previously misplaced (as detailed in other points below). 

• Materials and Methods: The authors should tailor this section (with subheadings and better organization) to specifically reflect how the materials and the methodology are designed to evaluate those hypotheses. Some results of the fragmentation analyses are included in this section and should be moved to the results (along with the ref to Table S1).

The Materials and Method section has been reorganized accordingly, adding subheadings that group the three levels of analysis – spatial (“Excavation”), bioanthropological (“Bioanthropological analysis”), and fragmentation (“Fragmentation analysis”). As suggested, parts pertaining to the results of the fragmentation analysis have been moved accordingly.

• The fragmentation analysis can be better presented and some references on the relative durability of certain skeletal components (femoral diaphysis, mandible corpus, etc. should be included here instead of the comment on selective representation of the cranium.

We have rephrased and added references, as follows: “Certain skeletal elements such as diaphyses of long bones, mandible corpus, parts of the temporal bones comprising the petrous process and the mastoid, thick cranial bones at the parietal and occipital level, and unerupted teeth still included in the maxillary bones are all more resistant to destruction from combustion at high temperatures [3,18,31,32]”.

• The use of the C index is fine, but calling it a ‘cranial index’ is not appropriate as a specific cranial index already exists. Is it not a cranial/postcranial index?

Thank you for pointing that out. Indeed, we have renamed the “cranial/postcranial index” abbreviated CPC index.

• The last paragraph of the M & M on p. 10 needs to be included with the results.

Done.

• Archaeological context: I have difficulty with the terminology in this section. The layer of major concern is US11, but the authors refer to US14 and 18 as concentrations of sherds within that layer. Are these then features, and not layers?

We have revised the terminology that was confused, as rightly pointed by the reviewer. In the Materials and Method section, we added a section related to the “Excavation” methods to better explain the coding. We changed the description of US 11 and other US as “features” throughout the text (including “The archaeological context” section and “The spatial analysis” of the Results section).

• Some of the information in this section is not concise or belongs in the discussion as interpretation of the behaviors leading to the deposits is offered.

Thank you for the suggestion. We have moved and re-arranged these parts in the Discussion section.

• I get the impression that the authors could use the spatial information more effectively for the analysis of Salorno as the comparison with other sites to give a better impression of the site size and density that may reflect on its formation.

Unfortunately, there are no coeval ustrina from the same/surrounding region that have been analyzed in depth. In the Introduction we refer to such lack of evidence and cite the few references available thus far: “[…] archaeological documentation of funeral pyres (or ustrina) is scarce and patchy [e.g. 1–7]. Such lack of evidence is not surprising if we think that ancient cremations were operated outdoors, by means of pyres that were seldom equipped with permanent or semi-permanent structures. As confirmed by a number of experimental archaeological studies, funeral pyres are extremely ephemeral in nature [1,8–13].”

• Results: Needs subheadings. Begin with the spatial analysis results on p. 12 and the paragraph mentioned above in the M & M. Much of this section is interpretation and not results. Separate out the results and discussion the interpretation in the discussion. This includes the comparisons with other sites, unless you did a specific analysis of the samples.

We have restructured the Results section entirely, adding subheadings that group results by type of analysis performed – ‘Bioanthropological analysis’, ‘Fragmentation analysis’, ‘Spatial analysis’. We have moved interpretative parts of the results in the Discussion section, along with comparative analyses with other sites. 

• Remove the two hypotheses section from p. 12.

Done.

• Identification and fragmentation: Shouldn’t the results of the two indices be here? Not in the discussion, where you contextualize them relative to other sites???

We have amended accordingly and consolidated information pertaining to the results in the Results section. 

• P. 11 line 255-256 seems to confuse use of maxillary for the upper jaw and mandible for the lower. The last line (261-262) of this paragraph makes no sense.

Checked and amended.

• Given what can be identified, which is what % of the total fragment weight, what would be the MNI and age and sex distribution using conventional procedures?

Thank you for the remark. Indeed, in the Results section ‘Bioanthropological analysis’ we have added a better description of the proportional weights of the cranial and post-cranial cremains, providing also for a quantification of the few identifiable fragments that have been used for calculating the MNI: “The minimum number of individuals (MNI) calculated with conventional procedures would be of four individuals: two adults, identified by a minimum number of two right mastoid processes and two right mandibular condyles; one child of about 6 years, identified by one right deciduous second molar and one left upper canine germ, both attributable to the same developmental stage (ca. 6 years ± 24 months) [48,49]; and one juvenile individual of 12 -14 years of age, identified by one right subadult head of radius. Obviously, the MNI thus calculated is evidently under-represented when compared with the total weight of human cremains found at the site”.

• Some interpretation of the fragmentation analysis is given in the Results, but this should be in the discussion.

The interpretative part of the fragmentation analysis has been moved to the Discussion as suggested.

• Discussion: Reorganize to include information moved from earlier sections. The table with the other sites is not needed and can be stated in the text. Be clear about why estimates for males and females are from different sites. The last section of the discussion, beginning with line 436, is good.

The Discussion section has been reorganized and subheadings have been added to better grouping interpretations of data by question: ‘Simulating a minimum number of individuals’; ‘Inferences from the spatial distribution of the human cremains’; ‘Cremation rituals at Salorno: some hypotheses’ with a 3rd level of heading ‘Salorno in the context of the Brandopferplätze’. 

Substantial parts have been moved from the Results and reorganized, as to center the discussion around the main research focus of the paper, i.e. whether the burnt deposits at Salorno show a primary context with no collection of cremains (hypothesis A), or whether they are the result of residuals from cremations after collection for internment (hypothesis B). The new section is broken down in three subheadings to discuss different aspects of the discussion – the MNI of individuals that we can infer from the amount of human cremains at Salorno; some inferences about the spatial distribution of the cremains; how we can use these data to propose some hypotheses about rituals during cremations at Salorno, and how to interpret them in the context of the Brandopferplätze – as the closest comparative reference for interpreting Salorno in a broader regional context. 

• References: need to be standardized to follow PLOS ONE formatting; Very inconsistent.

Done.

• Images: Is it possible to provide a map that includes the location of Salorno and the other sites used for comparison? 

Yes, we changed figure 1 accordingly.

• The view of the valley is not very informative. Delete

We removed figure 2 and put it in the Supplementary material.

• The images of bone condition and fragmentation, in situ and after sorting, are good; the images of the artifacts are less pertinent to the analysis presented here and most can be omitted. An image of the ritual cups is good as those are part of the material considered.

We moved the archaeological material figures to the Supplementary information.

---

## [Editor Report · Decision Letter 1]

11 Apr 2022

Salorno – Dos de la Forca (Adige Valley, Northern Italy): a unique cremation site of the Late Bronze Age.

PONE-D-21-34557R1

Dear Dr. Tecchiati,

We’re pleased to inform you that your manuscript has been judged scientifically suitable for publication and will be formally accepted for publication once it meets all outstanding technical requirements.

Kind regards,

Lynne A Schepartz

Academic Editor

PLOS ONE
---

## [Editor Report · Acceptance letter]

14 Apr 2022

PONE-D-21-34557R1 

Salorno – Dos de la Forca (Adige Valley, Northern Italy): a unique cremation site of the Late Bronze Age 

Dear Dr. Tecchiati:

I'm pleased to inform you that your manuscript has been deemed suitable for publication in PLOS ONE. Congratulations! Your manuscript is now with our production department. 

Kind regards, 

on behalf of

Dr. Lynne A Schepartz 

Academic Editor

PLOS ONE